# Divalent Yb-Doped Silica Glass and Fiber with High Quantum Efficiency for White Light Source

**DOI:** 10.3390/ma15093148

**Published:** 2022-04-26

**Authors:** Changming Xia, Jiantao Liu, Zhiyun Hou, Guiyao Zhou

**Affiliations:** 1Guangdong Province Key Laboratory of Nano-Photonic Functional Materials and Devices, South China Normal University, Guangzhou 510006, China; houzhiyun@163.com (Z.H.); gyzhou@scnu.edu.cn (G.Z.); 2Guangzhou Key Laboratory for Special Fiber Photonic Devices and Applications, South China Normal University, Guangzhou 510006, China; bzliujiantao@126.com

**Keywords:** Yb^2+^-doped glass, fiber laser, visible–infrared lasers, white light source, rod in tube

## Abstract

The 4f^13^5d–4f^14^ energy transition of Yb^2+^ ions can cover the whole white light wavelength, Yb^2+^-doped materials have thus been a hot research field. In order to obtain a white light source, many kinds of Yb^2+^-doped materials have been prepared. In this study, divalent Yb^2+^-doped silica fiber was fabricated using rod-in-tube technology. The fiber core of Yb^2+^-doped silica glass was prepared with high-temperature melting technology under vacuum conditions. The spectroscopic properties of the Yb^2+^-doped glass and fiber were studied. The experiments indicate that divalent Yb^2+^-doped glass has a high quantum efficiency and super-broadband fluorescence in the visible region with an excitation wavelength of 405 nm. In addition, the results suggest that Yb^2+^-doped fiber has a potential for application in visible fiber lasers and fiber amplification.

## 1. Introduction

In recent years, divalent Yb^2+^ ions have attracted considerable interest [1] because they provide a photodarkening effect, the exist of which is a key limitation of high-power fiber lasers [2,3]. Moreover, divalent Yb^2+^ ions have potential applications as white light sources [4]. The optical properties of Yb^2+^ ions in crystalline materials, such as CaF_2_ [5,6], SrI_2_ [7], MgF_2_ [8], YAG (Y_3_Al_5_O_12_, YAlO_3_) [9], NaI [10], phosphate crystals [11] and LiBaF_3_ [9] have been widely reported. In addition, the optical properties of Yb^2+^ ions in materials such as aluminosilicate glass [12] and silica glass [13,14] have also been widely investigated. Wang Chao reported upconversion luminescence of Yb^2+^/Yb^3+^ co-doped silica glass with femtosecond laser irradiation [15]. Xia Chang Ming et al. reported an influence of Yb^2+^ ions on the optical properties of Yb^3+^/Ho^3+^ co-doped silica glass synthesized through the non-chemical vapor deposition method [16]. Moreover, Yb^2+^ ions as scintillation sensitizers have also been reported [17]. The scintillation properties of Yb^2+^ ions in SrI_2_ [18], SrCl_2_, Cs_4_CaI_6_ and Cs_4_SrI_6_ [19] have been reported. Zhang Jiaqi et al. reported the photoluminescence properties of Yb^2+^-doped SrSi_2_O_2_N_2_ phosphor for image storage applications [20]. Ma Zhidong et al. reported the red persistent luminescence of Na_2_CaGe_2_O_6_: Yb^2+^ phosphor [21]. However, the laser performance of Yb^2+^ ions has not been investigated because of the high excited-state absorption of these ions [22,23].

The ground state configuration of Yb^2+^ ions is 4f^14^, whose excited state configuration is 4f^13^5d^1^ and consists of 140 energy levels. In addition, the state level of 4f^13^6s^1^ is the lowest excited energy level of the Yb^2+^ ions. However, in most materials, the 4f^13^6s^1^ energy level is lower than the 4f^13^5d^1^ level; therefore, there is excited-state absorption, which hinders laser oscillation. This is a limitation for the laser development of the Yb^2+^-doped material. Therefore, a material with the lowest excited level that belongs to the 4f^13^5d^1^ configuration is necessary. S. Kück believes that materials with a large crystal field can be used [1]. Here, the 4f^13^5d^1^ level largely splits, resulting in a 4f^13^5d^1^ level lower than the 4f^13^6s^1^ level.

In this study, the divalent Yb^2+^-doped silica fiber was fabricated by rod-in-tube technology. The fiber core of Yb^2+^-doped silica glass was prepared by high-temperature melting technology under vacuum conditions. The spectroscopic properties of the Yb^2+^-doped glass and fiber were studied. The laser performances of the divalent Yb^2+^-doped fibers were measured.

## 2. Experimental

The Yb^2+^-doped fiber was fabricated with rod-in-tube technology. The fiber core was Yb^2+^-doped silica glass with a composition of 97.95SiO_2_-1.86Al_2_O_3_-0.19Yb_2_O_3_ (mol%) (SiO_2_, Sinopharm Chemical Reagent Co., Ltd., Shanghai, China, 99.99%; Al_2_O_3_, Alfa 99.995%; Yb_2_O_3_, Alfa 99.99%). It was prepared through the traditional melting technology under the high temperature of 1800 °C using a graphite crucible. Graphite has intensity reduction properties under high temperature, which can promote the reduction of trivalent Yb^3+^ to divalent Yb^2+^ ions [4,24]. The absorption spectrum was recorded with an ocean optical spectrophotometer Maya 2000 (Ocean optics, Dunedin, FL, USA) in the range of 200–1100 nm, and a light source (DH-2000-BAL, Ocean optics, Dunedin, FL, USA) was used. The excitation spectrum with the emission centered at 541 nm and the emission spectrum under the excitation of 365, 405 and 427 nm were measured with an Edinburgh FL-FS 920 TCSPC (Edinburgh Instruments, Livingston, UK), in which an Xenon lamp in the range of 200–850 nm was used as the excitation source. The fluorescence lifetime was measured using the Edinburgh FL-FS 920 TCSPC (Edinburgh Instruments, Livingston, UK) [25,26]. The quantum efficiency was recorded with a commercialized system (XPQY-EQE-350-1100, Guangzhou Xi Pu Optoelectronics Technology Co., Ltd., Guangzhou, China) [27]. The refractive index profile of the Yb^2+^-doped fiber was measured with a digital holographic system (3D Refractive Index Measurements, Shanghai University, Shanghai, China) [28].

## 3. Results and Discussion

The Yb^2+^-doped silica glass sample, shown in Figure 1, presented a brownish yellow coloring, which can be attributed to the characteristic absorption of Yb^2+^ ions in the ultraviolet (UV) to visible spectral range, which is in agreement with [8].

The UV–visible–near-infrared absorption spectra (blue line) of Yb^2+^-doped silica glass and photoluminescence spectra (red line) with an excitation of 365 nm are shown in Figure 2a. The Yb^2+^-doped silica glass had intensity absorption in the UV–visible region, and this absorption band extended to the near-infrared region. In addition, there was a small absorption band at 976 nm corresponding to the characteristic f–f band transition of Yb^3+^ ions, which suggests the presence of few Yb^3+^ ions. A broad emission band centered at 542 nm was observed when it was excited with a wavelength of 365 nm in response to the 4f^13^5d–4f^14^ transition [29]. The band width was up to 155 nm, which suggests that Yb^2+^-doped silica glasses have outstanding optical properties. The Yb^2+^-doped silica glass exhibited photoluminescence quantum yields (QY) of 68.8% and 69.2% with excitations of 365 and 405 nm, respectively, which suggests a potential for use as a visible laser. Figure 3a shows the excitation spectrum with an emission wavelength of 540 nm, and Figure 3b shows the emission spectra with excitation wavelengths of 365, 405 and 427 nm for the Yb^2+^-doped silica glass. In addition, the direct and indirect optical band gap energy values of Yb^2+^-doped silica were calculated using the UV–visible absorption spectrum according to the Kubelka–Munk (K–M) theory [30,31]. The direct and indirect optical band gap energy values were 2.61 eV and 1.85 eV, respectively. The direct and indirect optical band gap energy spectrum is shown in Figure 2b,c.

From Figure 3a, there are three excitation wavelengths centered at 282, 365 and 434 nm corresponding to the 5d–4f^13^5d transition [29], and the excitation intensity of 434 nm was the largest. In Figure 3b, the emission peak was centered at 543, 534 and 541 nm when the excitation wavelength was located at 365, 405 and 427 nm, respectively. The emission bandwidths for the excitation wavelengths of 365, 405 and 427 nm were 209, 163 and 159 nm, respectively, which correspond to the double of the values reported in [8]. As seen from the Figure 3c chromaticity diagram, upon the excited wavelength of 365, 405 and 427 nm, the luminescence region located near the yellow region of the chromaticity diagram, the CIE1931 coordinates were (0.37, 0.45), (0.37, 0.48) and (0.41, 0.53), respectively.

The lifetime of Yb^2+^-doped silica glass was monitored at 540, 530 and 520 nm and adjusted to single exponential functions, as shown in Figure 4. The lifetimes were 78.048, 74.582 and 74.569 μs monitored at 540, 530 and 520 nm when the excitation wavelengths were 365, 405 and 427 nm, respectively. Using the lifetimes and quantum efficiency, the emission cross-section $\sigma_{\mathrm{em}}$ can be obtained with the McCumber formula [8]:(1)$$\sigma_{\mathrm{em}}=\eta \sqrt{\frac{\ln2}{\pi }}\frac{1}{4{\pi \mathrm{cn}}^{2}\tau }\frac{\lambda^{4}}{\Delta \lambda }$$
where η is the quantum efficiency, c is the speed of light, n is the refractive index, τ is the emission lifetime, λ is the peak emission wavelength and Δλ is the emission bandwidth. According to the results, the emission cross-sections of the excitation wavelength at 365 and 405 nm were 2.13825 × 10^−19^ and 2.69916 × 10^−23^ cm^2^, respectively, which are higher than those of Yb^2+^:MgF_2_ but smaller than those of Yb^2+^:KMgF_3_ as reported in [8].

Yb^2+^-doped glass with a thickness of 3 mm was used for a laser with 405 nm pumping. The surface of the glass was optically polished. A simple laser with an F–P cavity was constructed with two dichroic mirrors with high reflectivity (>99% and 85%) at 561 nm, as shown in Figure 5. However, instead of the laser, a super-broadband fluorescence in the visible region from 445 to 800 nm was obtained, with a bandwidth of more than 200 nm. Figure 5c shows the emission spectra of glass with different pump powers. The intensity of the spectra increased with increasing pump laser intensity. Figure 6 shows a photograph of a harsh white light emitted from the glass with an excitation wavelength of 405 nm.

The structure of the Yb^2+^-doped fiber fabricated with rod-in-tube technology is shown in Figure 7. The fiber core was made of Yb^2+^-doped silica glass, as presented in Figure 1, and the fiber and fiber core diameters were 156.06 and 27 μm, respectively. The refractive index profile of the Yb^2+^-doped fiber is shown in Figure 7b. The black and red lines represent the horizontal and vertical refractive index profiles, respectively. The refractive index of the immersion liquid was 1.462. The Δn between the fiber core and the fiber cladding was 4.26 × 10^−4^, whereas it was 2.737 × 10^−4^ in the fiber core.

Figure 8 shows the emission spectra of Yb^2+^-doped fiber with a length of 30 cm and excitations of 365 and 405 nm as a function of pump power: 100 and 150 mW for the 365 nm laser, and 100, 200, 400 and 600 mW for the 405 nm laser. Broad emission bands centered at 561 nm were observed with the excitation of 365 nm, which is attributed to the typical 4f^13^5d^1^–4f ^14^ transitions of Yb^2+^ ions [32]. The emission band covered wavelengths from 500 to 650 nm, and the full width at half maximum (FHWH) of the emission band was up to 82 nm. This suggests that the Yb^2+^-doped fiber can be used to produce tunable fiber lasers operating in the visible region. When the excitation wavelength was 405 nm, broad emission bands centered at 631 nm were also observed. The emission band covered wavelengths from 550 to 850 nm, and the FHWH of the emission band was up to 91 nm. These results suggest that Yb^2+^-doped silica fiber can be used to produce tunable fiber lasers operating in the visible region.

The emission spectra of Yb^2+^-doped fiber with lengths of 30 and 60 cm as a function of pump power at the excitation of 405 nm are shown in Figure 9. As seen, the emission spectra for different lengths were different. When the length of the fiber was short, there were two emission peaks centered at 586 and 620 nm. The FHWH of the emission band was up to 108 nm, and the wide emission bands suggest that Yb^2+^-doped fiber can be used to produce a tunable fiber laser operating in the visible region. When the fiber length was longer, because of the reabsorption of fluorescence, the centered wavelength of the emission band shifted to 626 nm, which is in agreement with the results in [4]; as the fiber length increased, the whole emission band width was narrow, and there was only an emission peak around 620 nm, which is ascribed to the reabsorption of the fluorescence long optical fiber. In addition, with the increase in pump power, the emission-band-centered wavelength also shifted to longer wavelengths.

To study the laser performance of the Yb^2+^-doped fiber, a free-space laser resonator cavity was constructed with an Yb^2+^-doped fiber of 30 cm. The laser experiment setup is shown in Figure 10. The cavity was constructed with two dichroic mirrors with high reflectivity (>99% and 85%) at 561 nm. The pump source was a 405 nm laser diode with a laser spot diameter of 105 μm and a numerical aperture of 0.22. The pump light was collimated and focused into the inner cladding of the fiber by an 8X objective lens. Figure 11 shows the change in the emission spectra as a function of the angle between the fiber face and the laser cavity mirror. As shown in Figure 11, with a reduction in the angle, the intensity of the emission spectra increased. Two emission peaks centered at 635 and 679 nm were observed. Moreover, as the angle increased, the emission peak centered at 679 nm disappeared, and only the emission peak centered at 635 nm was observed, which suggests that a free-space laser resonator cavity was reformed. Figure 12 shows the intense emission peak centered at 637 nm as a function of pump power with an excitation of 405 nm. However, because the available pumping power was limited, a remarkable laser emission could not be observed. The results of the experiments suggest that the Yb^2+^-doped fiber laser operating at the visible wavelength is feasible. In future works, more optical experiments on the Yb^2+^-doped fiber will be performed for the fiber laser.

## 4. Conclusions

A divalent Yb^2+^-doped silica fiber for a white light source was fabricated through the rod-in-tube method. Yb^2+^-doped silica glass for the fiber core was prepared with high-temperature melting technology under vacuum conditions. The spectroscopic properties of Yb^2+^-doped glass and fiber were studied. An intense emission peak centered at 637 nm was observed in the divalent Yb^2+^-doped fiber. The results show that the divalent Yb^2+^-doped glass has a high quantum efficiency and super-broadband fluorescence in the visible region with an excitation wavelength of 405 nm. The laser experiment for Yb^2+^-doped silica fiber was performed; however, because the available pumping power was limited, a remarkable laser emission could not be observed. In future works, more optical experiments on the Yb^2+^-doped fiber will be performed in order to obtain the fiber laser. Based on the above results, these experiments suggest that Yb^2+^-doped silica and fiber have potential applications in visible fiber lasers and amplification.

## Figures and Tables

**Figure 1 materials-15-03148-f001:**
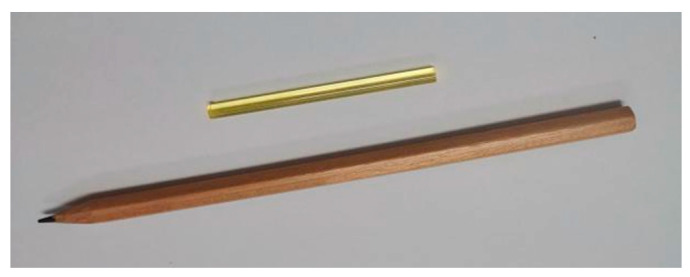
Yb^2+^-doped silica glass samples.

**Figure 2 materials-15-03148-f002:**
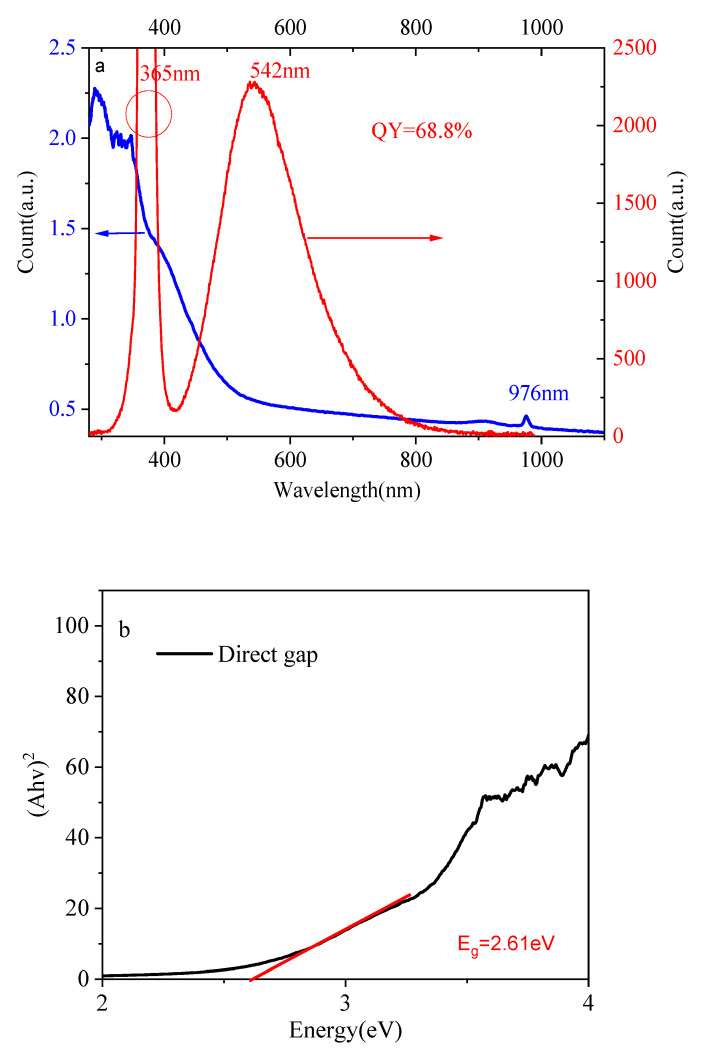
(**a**) UV–near-infrared absorption and photoluminescence spectra with 365 nm excitation; (**b**) direct optical band gap energy spectrum; (**c**) indirect optical band gap energy spectrum.

**Figure 3 materials-15-03148-f003:**
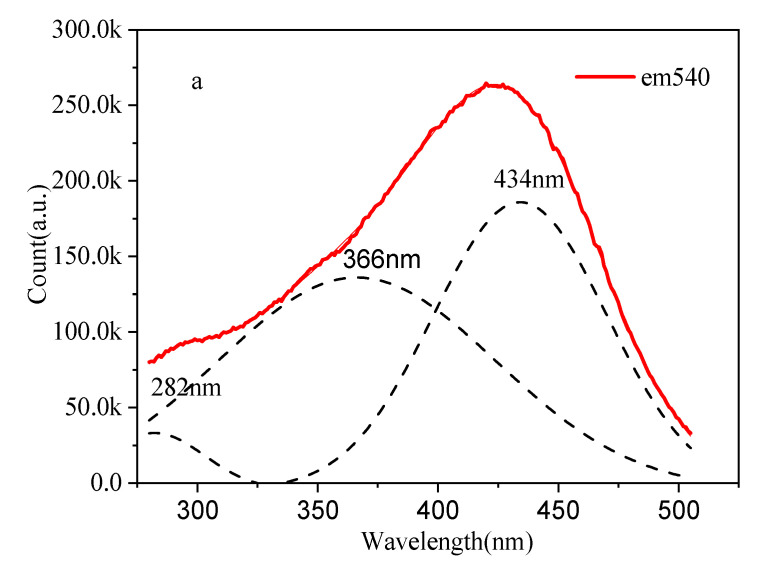
Excitation and emission spectra of Yb^2+^-doped silica glass. (**a**) Excitation spectrum with an emission wavelength of 540 nm; (**b**) emission spectra with excitation wavelengths of 365, 405 and 427 nm; (**c**) chromaticity diagram for various excited wavelengths.

**Figure 4 materials-15-03148-f004:**
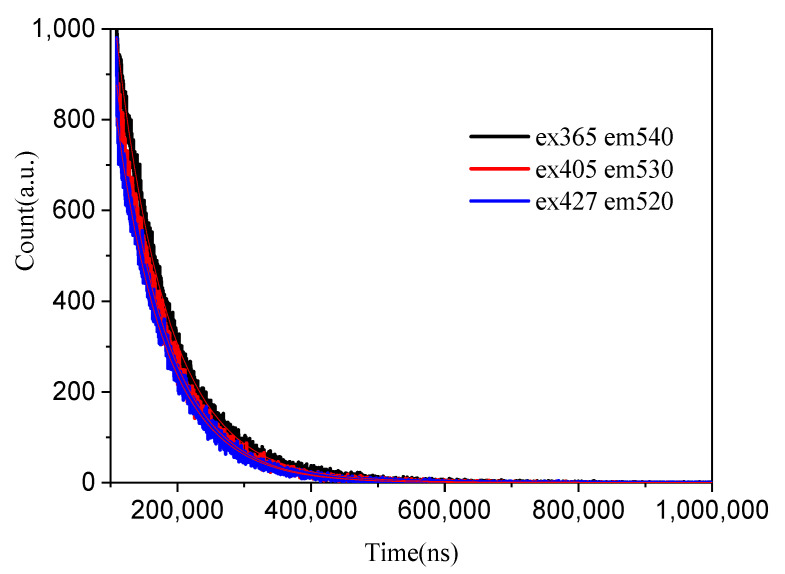
Lifetimes of Yb^2+^-doped silica glass monitored at 540, 530 and 520 nm when the excitation wavelength was 365, 405, and 427 nm.

**Figure 5 materials-15-03148-f005:**
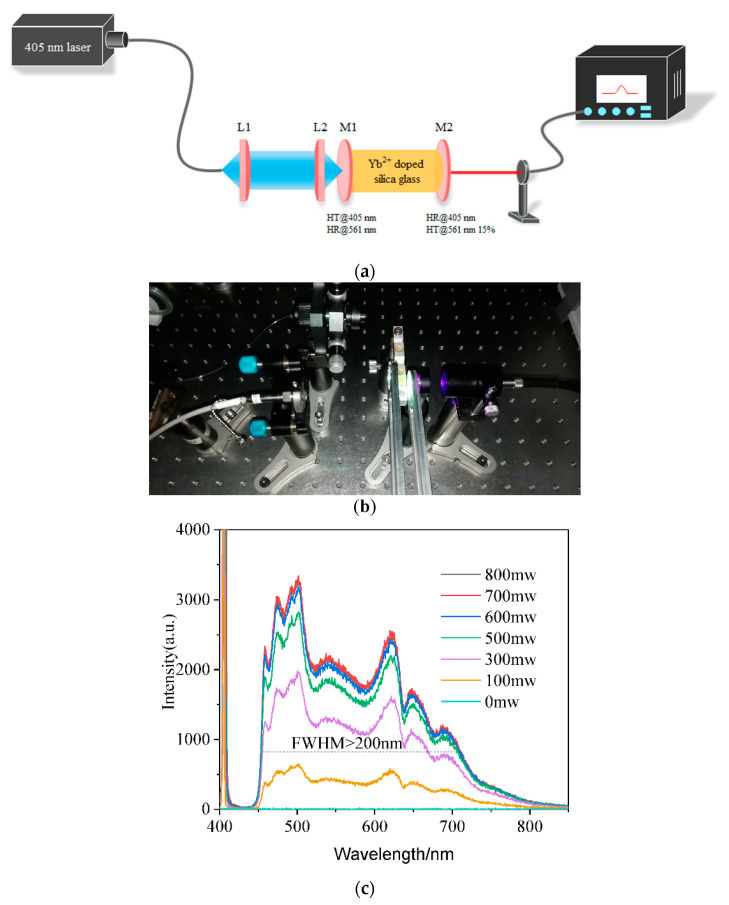
(**a**) Scheme of Yb^2+^-doped glass laser; (**b**) a simple glass solid laser cavity with Yb^2+^ glass and (**c**) super-broadband fluorescence of Yb^2+^-doped glass in the F–P cavity using a 405 nm laser pump.

**Figure 6 materials-15-03148-f006:**
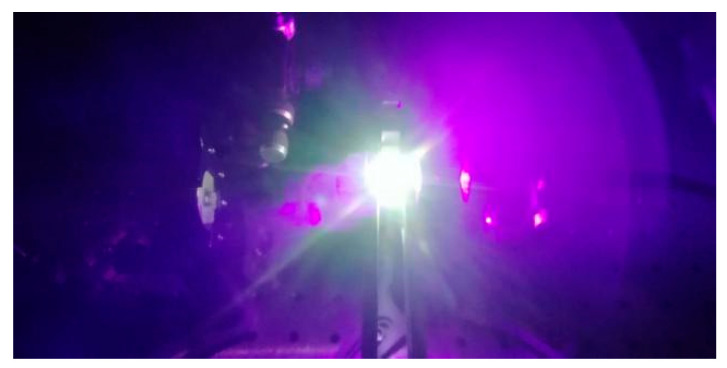
Harsh white light photograph of Yb^2+^-doped glass with an excitation of 405 nm.

**Figure 7 materials-15-03148-f007:**
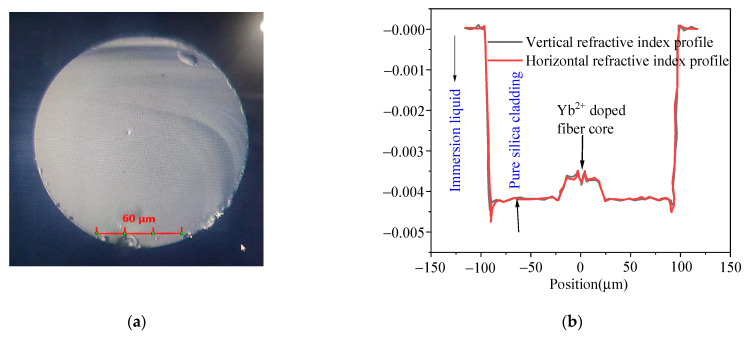
(**a**) Typical scanning electron microscopy image (scale bar of 30 μm) and (**b**) refractive index profile of the Yb^2+^-doped fiber.

**Figure 8 materials-15-03148-f008:**
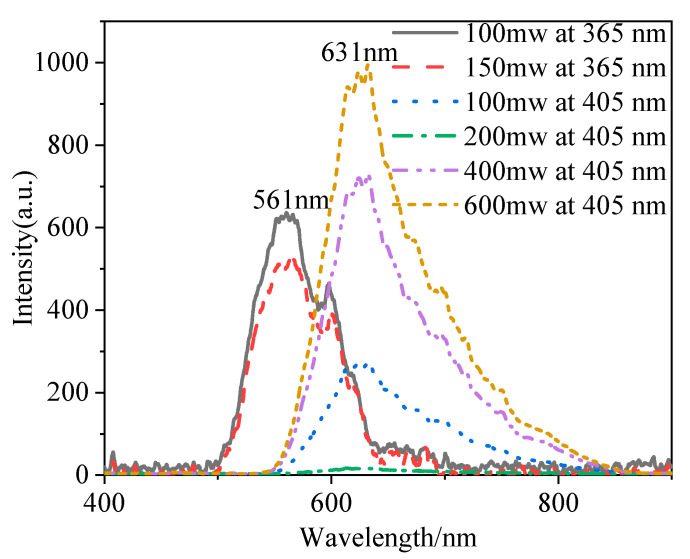
Emission spectra of Yb^2+^-doped fiber with a length of 30 cm as a function of pump power with excitations of 365 and 405 nm.

**Figure 9 materials-15-03148-f009:**
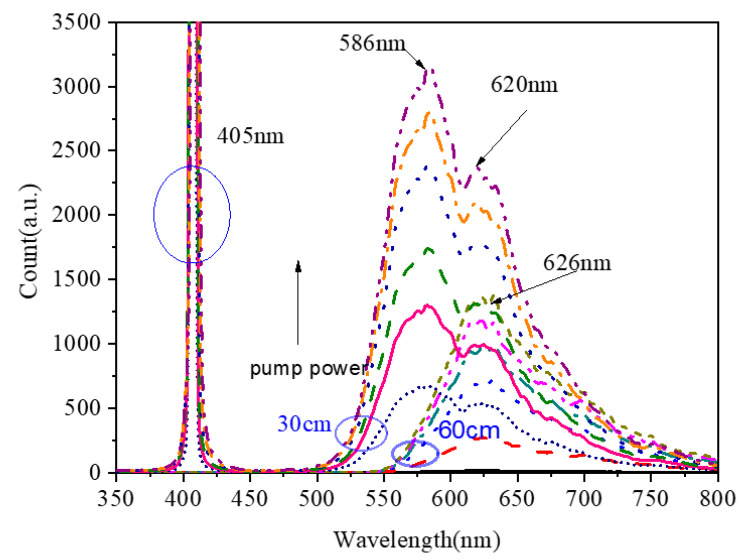
Emission spectra of Yb^2+^-doped fiber lengths of 30 and 60 cm as a function of pump power at the excitation of 405 nm.

**Figure 10 materials-15-03148-f010:**
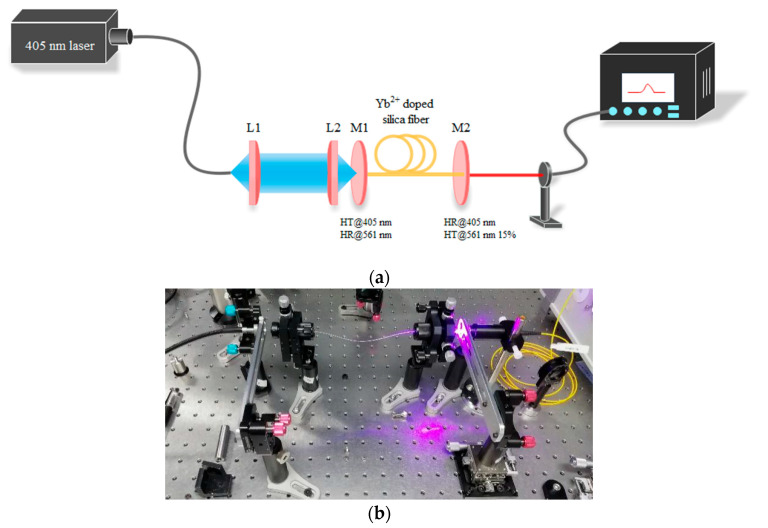
(**a**) Scheme and (**b**) setup of the Yb^2+^-doped fiber laser.

**Figure 11 materials-15-03148-f011:**
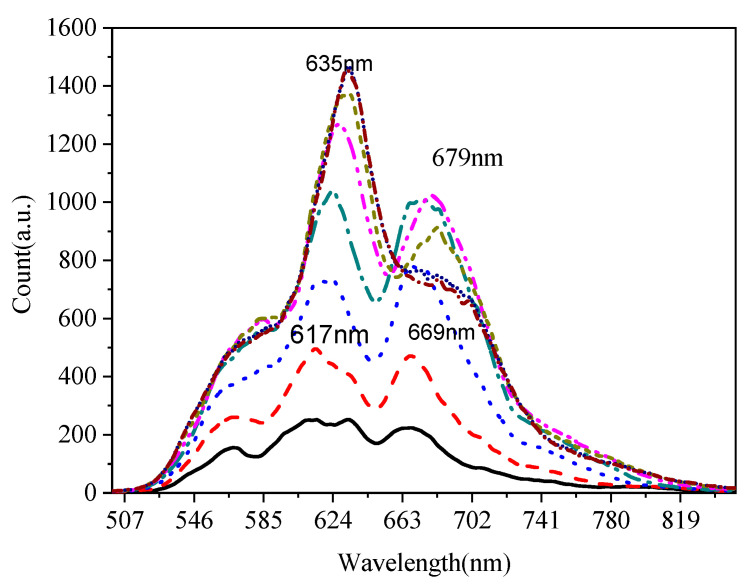
Emission spectra of the Yb^2+^-doped silica fiber as a function of the angle between fiber face and laser cavity mirror at an excitation of 405 nm.

**Figure 12 materials-15-03148-f012:**
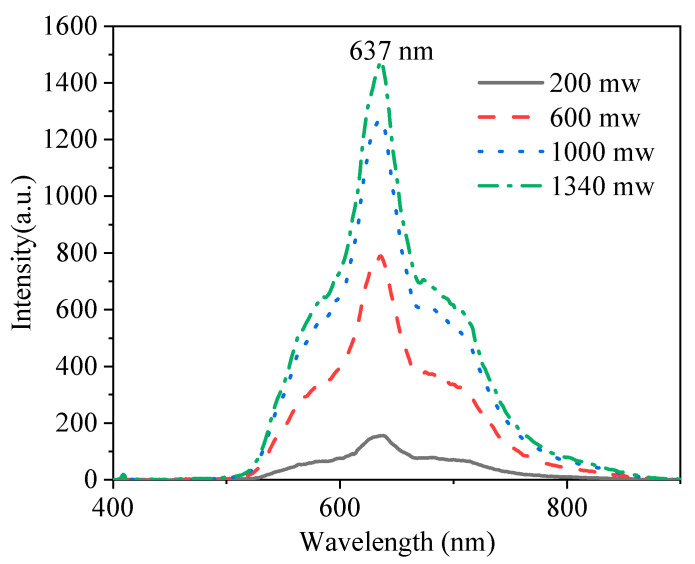
Emission spectra of the Yb^2+^-doped silica fiber as a function of pump power at an excitation of 405 nm.

## Data Availability

All data, models, and code generated or used during the study appear in the submitted article.

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
