# Peer review of "Divalent Yb-Doped Silica Glass and Fiber with High Quantum Efficiency for White Light Source"

_materials, 2022, doi:10.3390/ma15093148_

Round 1
Reviewer 1 Report
The article Divalent Yb doped silica glass and fiber with high quantum efficiency for white light source is devoted to the fabrication of Yb2+ doped silica fibers. Undoubtedly, the results presented by the authors are of high scientific novelty and practical significance, and are also promising for practical research. In general, the presented results of the study can be accepted for publication after the authors provide answers to all the questions raised by the reviewer during the reading of the article.
1. In the abstract, the authors need to more clearly state the purpose and relevance of this work.
2. In the abstract, the authors should explain what caused the choice of Yb2+ as a dopant, as well as the method for obtaining quartz fibers.
3. The authors should describe in more detail the effect of changing the width of the spectral lines on the properties of the fibers, as well as the effect of the dopant.
4. The authors should explain how homogeneous the distribution of the Yb2+ dopant in the composition of the quartz fiber is and whether this distribution affects the spectroscopic properties of the fibers.
Author Response
Dear reviewer,
Thank you again, thank you very much for giving us a more chances to revise our paper to improve the quality of papers. We carefully answered the comments of each reviewer and marked them in the original text

Reviewer 2 Report
In this study, the fiber core of Yb2+-doped silica glass was manufactured by high-temperature melting technique under vacuum circumstances employing rod-in-tube technology. The characteristics of Yb2+-doped glass and fiber were investigated spectroscopically. The findings indicate that Yb2+-doped fiber could be used in visual fiber lasers and fiber amplification. Through the review, I think this is a very nicely written paper, and the author analyzed and discussed the results in detail and correctly. The analysis developed in this paper is correct and the obtained results are interesting. The paper has sufficient novelty which covers the scope of the Materials journal. Therefore, the manuscript may consider for publication in the Materials journal after responding to the following comments and major revising the manuscript properly.
- In the last paragraph of the Introduction section author mentioned some of the obtained results of this study. For example: "The experiments indicate that the Yb2+-doped glass has a high quantum efficiency, and Yb2+-doped fiber has a potential application in visible fiber lasers". Also "The results suggest that in the 48 future Yb2+-doped fiber can be widely applied in the development of fiber laser and fiber amplification operating in the visible region." Authors are suggested deleting these sentences from the Introduction.
- More recent relevant literature or similar work discussion is mandatory in the introduction section, which is missing in the Introduction. Authors are suggested to add one paragraph in the introduction section by discussing the recent progress citing similar work.
- Various materials, chemicals, and characterization devices are used in this experiment. Please write all of these materials and devices model, and origin/country clearly.
- Quality of the figures 2, 3, 5(a)&(c), 7(b), 8, 9, 10(a), 11, and 12 is not good enough.
- The number of references needs to be increased.
- More recent relevant references need to be included in the Introduction and Discussion sections.
- Authors are suggested to use more references from recent past, and recommend to cite following references in the introduction section. DOI: 10.1021/acsaelm.1c00682; 10.1016/j.optmat.2013.07.025; 10.1021/acsaelm.1c00703.
- Authors are suggested to calculate and include the direct and indirect optical band gap energies values and corresponding figures.
- Also if possible please include the chromaticity diagram for various wavelengths.
- Check the typos throughout the manuscript during revision submission.
Author Response

(The authors gave the same response as above.)

Round 2
Reviewer 1 Report
The authors answered all the questions, the article can be accepted for publication.
Reviewer 2 Report
The authors properly improved the manuscript based on my comments. Thank you very much for the revision.